

# Repeated ethanol exposure increases anxiety-like behaviour in zebrafish during withdrawal

Jeffrey T. Krook[1], Erika Duperreault[1], Dustin Newton[1], Matthew S. Ross[2] and Trevor J. Hamilton[1,3]

[1] Department of Psychology, MacEwan University, Edmonton, AB, Canada
[2] Department of Physical Sciences, MacEwan University, Edmonton, AB, Canada
[3] Neuroscience and Mental Health Institute, University of Alberta, Edmonton, AB, Canada

## ABSTRACT

Zebrafish (*Danio rerio*) are quickly becoming an important model organism in behavioural neuroscience and drug addiction research. Conditioned place preference studies show that drugs of abuse produce responses in zebrafish that are similar to mammalian animal models. Repeated administration of ethanol in zebrafish results in withdrawal-induced behavioural responses that vary with dose and exposure duration, requiring additional investigation. Here, we examine the effects of ethanol withdrawal on anxiety-like behaviours in adult zebrafish after a 21-day ethanol dosing schedule at either 0.4% or 0.8%. Anxiety-like behaviour was measured with the novel object approach test; this test involves placing a fish in a circular arena with a novel object in the centre and observing the amount of exploration of the object. We found increased anxiety-like behaviour during ethanol withdrawal. This study adds to the growing body of literature that validates the zebrafish as a model organism in the field of behavioural neuroscience and addiction.

## INTRODUCTION

Alcohol (ethanol) is amongst the most commonly abused legal substances worldwide (*World Health Organization, 2011*). Excessive and repeated alcohol use is known to cause significant impairments in behaviour and physiology. Many of the detriments of alcohol use come from repeated exposure due to an addiction called alcohol use disorder (*Levinthal & Hamilton, 2015*; *American Psychiatric Association, 2013*). Alcohol use disorder can be characterized by several symptoms: an impulse to seek out and consume alcohol, inhibition of controlled consumption, and the emergence of a negative emotional state during withdrawal (*Vendruscolo & Roberts, 2014*), as well as many negative physical manifestations leading to increased mortality (*Kendler et al., 2016*). Along with the severe health and behavioural detriments, alcohol abuse is also a societal and economic burden that is undertreated (*Holt & Tobin, 2018*). In part, the difficulty lies in the fact that individuals addicted to alcohol consumption that refrain from alcohol use will often

Corresponding author
Trevor J. Hamilton,
trevorjameshamilton@gmail.com

experience strong aversive effects of withdrawal, such as increased stress, motivating them to resume alcohol consumption (*Becker, 2012*).

In order to understand withdrawal behaviours, an understanding of the mechanism through which ethanol acts on the brain is needed. Acute ethanol administration increases the activity of the inhibitory neurotransmitter γ-Aminobutyric acid (GABA) at the $GABA_A$ receptor (*Faingold, N'Gouemo & Riaz, 1998*). Ethanol agonistically activates $GABA_A$ receptors, causing a net reduction on the excitability of neurons (*Gilpin & Koob, 2008*). Due to the inhibitory nature of $GABA_A$ receptors, the result of ethanol exposure in the central nervous system (CNS) is a reduction in anxiety-like symptoms (*Gilpin & Koob, 2008*). This ethanol-induced reduction in anxiety contributes to the addictive nature of ethanol. A further contributing factor to the addictive nature of ethanol is the upregulation of the dopaminergic system during ethanol exposure (*Chatterjee & Gerlai, 2009*). Dopamine release is associated with a pleasurable response, reinforcing the motivation to abuse ethanol. Further contributions to ethanol addiction come from the stimulation of the opioid system through endogenous opioids, as well as the release of serotonin and glutamate (*Gilpin & Koob, 2008*). However, it is neuroadaptation of the $GABA_A$ receptor to the continued presence of alcohol that results in the increased anxiety, delirium tremens and seizures experienced during alcohol withdrawal that make withdrawal a potentially lethal occurrence (*Silberman et al., 2009*).

The zebrafish (*Danio rerio*) is an ideal model organism for exploring the effects of alcoholism and withdrawal because of the ease of dosing large numbers of zebrafish in an identical manner, the sequenced genome for potential genetic manipulations (*Howe et al., 2013*), and the presence of analogous neurotransmitter systems to humans that mediate ethanol responses within the CNS (*Gerlai et al., 2009*). Studies on acute ethanol exposure in zebrafish demonstrate that ethanol has anxiolytic properties (*Mathur & Guo, 2011*; *Johnson & Hamilton, 2017*; *Hamilton et al., 2017*). Several of the reported behaviours associated with the reduction in anxiety-like behaviours due to acute (1%) ethanol exposure in zebrafish are: increased time swimming near the top of the tank, reduced jumping, more transitions to the top of the tank, and reduced anti-predatory responses (*Tran et al., 2016*). However, it is important to note that with longer durations of ethanol exposure (1 h) the anxiolytic effect on behaviour seems to change to an anxiogenic state (*Rosemberg et al., 2012*; *Mocelin et al., 2018*).

Studies examining chronic ethanol exposure in zebrafish have reported inconsistent results, possibly due to the many different exposures times, duration of exposure, duration of withdrawal, and concentration of ethanol used (*Da Silva Chaves et al., 2018*). To date there have been many studies examining the impact of chronic ethanol exposure (*Müller et al., 2017*; *Tran, Chatterjee & Gerlai, 2015*; *Wong et al., 2010*), and yet only a few that have examined withdrawal behaviour (*Holcombe et al., 2013*; *Cachat et al., 2010*, *Mathur & Guo, 2011*; *Tran et al., 2016*; *Pittman & Hylton, 2015*; *Gerlai et al., 2009*; *Pittman & Ichikawa, 2013*; *Müller et al., 2017*; *Dewari et al., 2016*; *Benneh et al., 2017*). For example, a study by *Müller et al. (2017)* examined the effects of repeated ethanol exposure over the course of 8 days, at 20 min per day, and a dose of 1% v/v. They found a significant increase in anxiety on the 9th day (1 day of withdrawal from ethanol) using a test of

shoaling behaviour in groups of zebrafish. In a similar study, *Cachat et al. (2010)* examined effects of ethanol withdrawal following 7 days of repeated ethanol dosing at a dose of 0.3% v/v for 20 min per day. Using whole body cortisol levels, which are known to correlate closely with withdrawal-evoked anxiety (*Keedwell et al., 2001*; *Nava et al., 2007*), the authors observed a significant elevation in cortisol levels during ethanol withdrawal (*Cachat et al., 2010*). Although the authors did not observe overt behavioural changes at the levels of ethanol exposure tested, it is likely there was an increase in stress and anxiety, as suggested by the increased cortisol levels (*Cachat et al., 2010*). This lack of behavioural change may be explained by the finding that withdrawal, although typically developing 6–24 h after cessation of ethanol consumption, may be delayed for up to 5 days (*Mathur & Guo, 2011*).

Tests of anxiety-like behaviour during ethanol withdrawal in zebrafish demonstrate increased anxiety in the novel tank diving test (*Mathur & Guo, 2011*; *Tran, Chatterjee & Gerlai, 2015*; *Pittman & Hylton, 2015*; *Cachat et al., 2010*; *Pittman & Ichikawa, 2013*; *Benneh et al., 2017*), increased anxiety in the shoaling test (*Gerlai et al., 2009*; *Müller et al., 2017*), and increased anxiety in the light/dark test (*Mathur & Guo, 2011*; *Pittman & Ichikawa, 2013*) (for review see *Da Silva Chaves et al., 2018*). However, previous research from our lab demonstrated increased light preference in a light/dark test during withdrawal after a 21-day exposure to repeated intermittent ethanol exposure (0.2%) (*Holcombe et al., 2013*). Usually increased light preference is associated with decreased anxiety which can be caused by anxiolytic drugs (*Maximino et al., 2014*). However, we suggested that this light preference during ethanol withdrawal was due to a conditioned-place preference as the fish were dosed with ethanol in a well-lit environment with white partitions around the dosing tanks that were similar to the white walls in the light/dark test, but this was not empirically tested.

To further investigate whether repeated ethanol exposure produces increased anxiety during withdrawal, in this study we exposed zebrafish to 1 h of ethanol exposure (0.4% or 0.8% v/v) per day, for 21 days, and then tested anxiety-like behaviour after 2 days of withdrawal. We used the novel object approach test, a validated test for anxiety in zebrafish (*Stewart et al., 2012*; *Johnson & Hamilton, 2017*, *Hamilton et al., 2017*) and measured anxiety-like behavioural responses and locomotion. Specifically, we measured: the time spent in different zones of the arena, immobility, and velocity of each fish during the novel object approach test.

## MATERIALS AND METHODS

### Animals and housing

Adult wild-type (short-fin) zebrafish of mixed sexes (~50:50 males and females, $n = 58$) were purchased from Aquatic Imports (Calgary, AB) and were quarantined for 60 days (same water quality parameters as below) prior to being moved to the habitat. Fish were randomly organized into groups of a maximum of 10 fish per 3 L polypropylene tank in a three-tier bench-top habitat system (Aquatic Habitats, Aquatic Ecosystems, Inc. Apopka, FL, USA). Habitat water was constantly filtered, aerated, maintained at between

26–28 °C, and controlled with non-iodized salt, sodium bicarbonate, and acetic acid. Husbandry was as described previously (*Holcombe et al., 2013*; *May et al., 2016*). Zebrafish were maintained on a 12 h light/12 h dark photoperiod, with lights on at 8:00 am and off at 8:00 pm (*Hamilton et al., 2016*; *May et al., 2016*; *Rimstad et al., 2017*; *Tsang et al., 2018*). Fish were fed fish pellets (Gemma Micro 300, Skretting by Nutreco, Westbrook, USA) and dry brine shrimp (Omega One Freeze Dried Mysis Shrimp nutri-treat; OmegaSea LLC, Sitka, USA) once per day, at least 30 min following dosing or behavioural testing. All experiments were approved by the MacEwan University Animal Research Ethics Board under protocol number 05-12-13, in compliance with the Canadian Council for Animal Care guidelines for the care and use of experimental animals. All fish were experimentally naïve.

## Repeated intermittent ethanol administration

The treatment groups consisted of fish exposed daily for 1 h to 0% (control), 0.4%, or 0.8% ethanol (vol/vol) for 21 consecutive days between the hours of 8:00 am. and 4:00 pm. The 21-day exposure period was based on a previous study conducted in our lab (*Holcombe et al., 2013*). Zebrafish were randomly netted from the large quarantine tank and assigned to one of two replicate tanks per condition with 9–10 fish per tank (~50:50 males and females). All groups were kept in separate 3 L tanks in the aquatic habitat, housed on the same tier (second from the top) in the middle of the row with adjacent tanks containing equal numbers of fish. Within each tank a spawning insert was used to transfer fish between habitat tanks to dosing tanks (*Holcombe, Schalomon & Hamilton, 2014*). Dosing tanks were prepared by adding ethanol directly to dosing tanks containing habitat water, to reach a concentration of either 0.4% or 0.8% ethanol in each dosing tank (*Holcombe, Schalomon & Hamilton, 2014*). After the dosing tanks were prepared the fish were transferred from the habitat tanks. Dosages were based on other studies using similar ethanol exposure times and concentrations (*Holcombe et al., 2013*; *Mathur & Guo, 2011*).

Daily ethanol exposure was set to 1 h, as internal ethanol concentrations approach a steady state within 60 min of exposure (*Tran, Chatterjee & Gerlai, 2015*). This method of intermittent repeated ethanol administration is preferred over a continuous ethanol exposure method, which involves administering ethanol directly to the habitat tanks of the test subjects for an extended period of time. The intermittent method allows greater translational relevance as it more closely mirrors the repeated and intermittent consumption patterns in humans (*Tran et al., 2016*; *Müller et al., 2017*). After dosing, zebrafish were transferred back to their habitat tanks. The control groups underwent an identical procedure, except the water they were transferred into consisted of only habitat water. Throughout the duration of dosing, water was maintained at a temperature of 26–28 °C using heating pads (Hydrofarm, Petaluma, USA). The dosing area was isolated using white corrugated plastic barriers throughout the dosing period to minimize any external visual stimuli. Behavioural testing was conducted 2 days after the last dose, allowing for 2 days of withdrawal between dosing and behavioural testing. Researchers were not blind to the treatment groups during dosing or testing.
a)

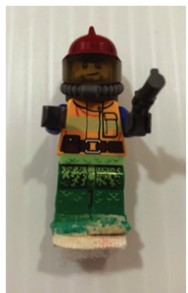

b)

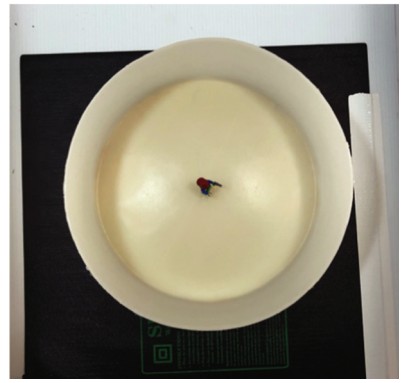

**Figure 1 Novel object approach test.** (A) The novel object used in the test was a multicolour LEGO figurine (height: five cm, width: 1.5 cm). (B) The novel object was placed in the centre of the circular arena (diameter: 34 cm, wall height: 16 cm).  

## Behavioural testing

The novel object approach test was used to measure locomotion and anxiety-like behaviour in individual zebrafish (Figs. 1A and 1B). The arena used was an opaque, white plastic cylinder (34 cm diameter, 15 cm walls) with a water depth of six cm (*Ou et al., 2015*; *Johnson & Hamilton, 2017*). The water was maintained at 26–28 °C using electric heating pads placed underneath the arena. Behavioural testing was conducted between 8:00 am and 4:00 pm. Fish were transferred from the main habitat to the adjacent experiment room by moving their 3 L polypropylene habitat tanks and placing them on heating mats in the experiment room, and allowing fish to acclimate to the testing room for a minimum of 10 min prior to behavioural testing. Following acclimation, subjects were individually netted and placed in the testing arena. Fish were released into the arena facing the novel object; the object for this test was a five cm tall Lego figurine (Fig. 1A; *Ou et al., 2015*; *Johnson & Hamilton, 2017*; *Hamilton et al., 2017*) and the location of the fish was recorded for a period of 15 min (*Hamilton, Holcombe & Tresguerres, 2014*). In order to record the movement of individual fish during behavioural trials, the differencing method on Ethovision XT (version 10; Noldus, Leesburg, VA, USA) motion tracking software was used. The dependent variables measured during the test were: time spent in the centre (near the object), transition, and thigmotaxis (near the wall) zones, as well as velocity (cm/s) and immobility, defined as the percent change in the pixels of the fish

from frame to frame, set at a threshold of 5% (*Pham et al., 2009*). Luminance in the arenas was measures by Cooke Cal-Spot 401 Calibrated precision spot photometer (Calright Instruments, San Diego, CA USA) and was 33 cd/m$^3$. Fish were euthanized immediately following behavioural testing by immersion into an MS-222 solution.

### Statistical analysis

Statistical significance was determined with an alpha of $P < 0.05$, and a 95% confidence interval. All data sets were assessed for normality using a D'Agostino and Pearson omnibus normality test. In order to examine differences between control and experimental groups parametric data was assessed using unpaired $t$-tests or one-way ANOVA with Bonferroni's multiple comparison post hoc tests. Data that were not normally distributed were examined using Kruskal–Wallis tests with Dunn's multiple comparison post hoc tests. In order to compare the two replicate groups from each prescribed ethanol exposure condition, we used $t$-tests and Mann–Whitney tests. We found no significant differences between groups, allowing us to pool the data. Behavioural data was analysed GraphPad Prism software (Version 6; GraphPad, San Diego, CA, USA). Data from fish with over 300 s of immobility were removed from the analysis (control, $n = 2$; 0.4%, $n = 4$; 0.8%, $n = 1$). All data are presented as mean ± S.E.M.

## RESULTS

### Repeated ethanol exposure: time in zones

In order to quantify the impact of ethanol-induced withdrawal on anxiety and exploratory behaviour, the time spent in the centre, transition, and thigmotaxis zones of the arena were quantified. Fish exposed to 0.8% ethanol spent less time in the centre zone of the arena, near the novel object (Fig. 2A; control: $9.7 \pm 2.3$ s, $n = 18$; 0.4%: $10.8 \pm 2.2$ s, $n = 16$; 0.8%: $2.8 \pm 0.62$ s, $n = 17$; $F(2, 48) = 5.089$, $P = 0.0099$). We did not observe a significant difference in time spent in each of the transition (Fig. 2B; control: $204.4 \pm 29.6$ s, $n = 18$; 0.4%: $216.2 \pm 31.4$ s, $n = 16$; 0.8%: $174.6 \pm 41.0$ s, $n = 17$; $H(3, 51) = 2.803$, $P = 0.2462$), or thigmotaxis (Fig. 2C; control: $685.7 \pm 30.8$ s, $n = 18$; 0.4%: $672.9 \pm 32.5$ s, $n = 16$; 0.8%: $722.5 \pm 40.9$ s, $n = 17$; $H(3, 51) = 3.217$, $P = 0.2002$) zones following ethanol exposure, at either 0.4% or 0.8% ethanol exposure conditions compared to controls.

### Repeated ethanol exposure: locomotion

To investigate any potential alterations of locomotor ability we quantified velocity and immobility. There were no significant differences in velocity (Fig. 3A; control: $8.7 \pm 0.4$ cm/s, $n = 18$; 0.4%: $9.1 \pm 0.8$ cm/s, $n = 16$; 0.8%: $9.8 \pm 0.8$ cm/s, $n = 17$; $H(3, 51) = 0.8916$, $P = 0.6403$) or immobility (Fig. 3B; control: $46.7 \pm 16.9$ s, $n = 18$; 0.4%: $30.19 \pm 13.7$ s, $n = 16$; 0.8%: $28.5 \pm 13.4$ s, $n = 17$; $H(3, 51) = 2.924$, $P = 0.2317$) between either treatment group and control fish.

## DISCUSSION

In the current study, we investigated whether repeated ethanol exposure alters anxiety-like behaviour using the novel object approach test. Our findings demonstrate that repeated ethanol exposure increases anxiety-like behaviour in zebrafish during withdrawal.

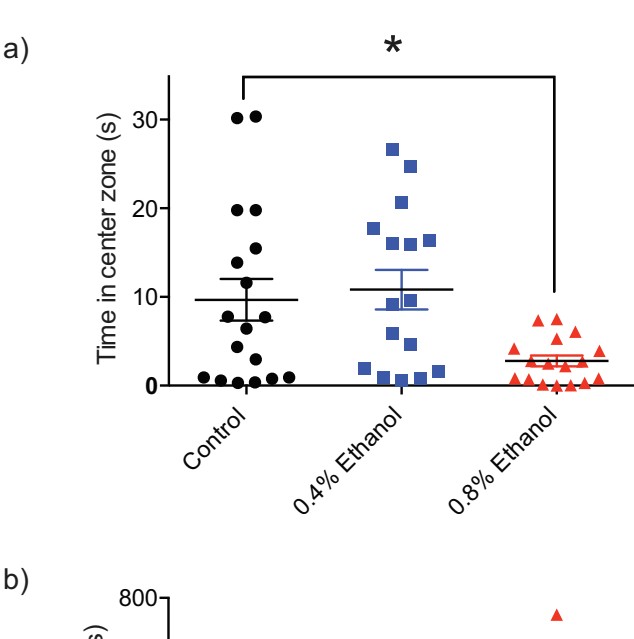

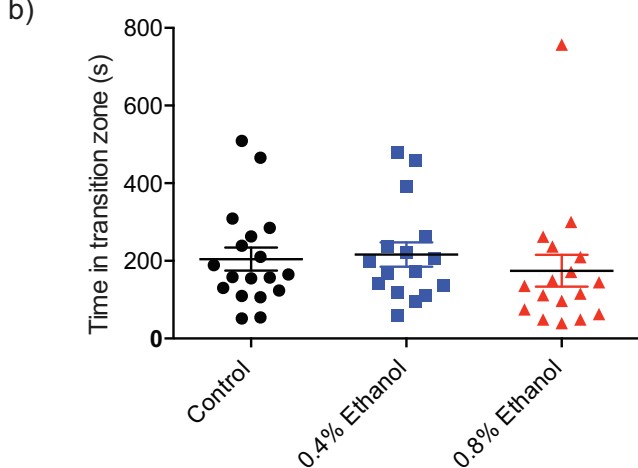

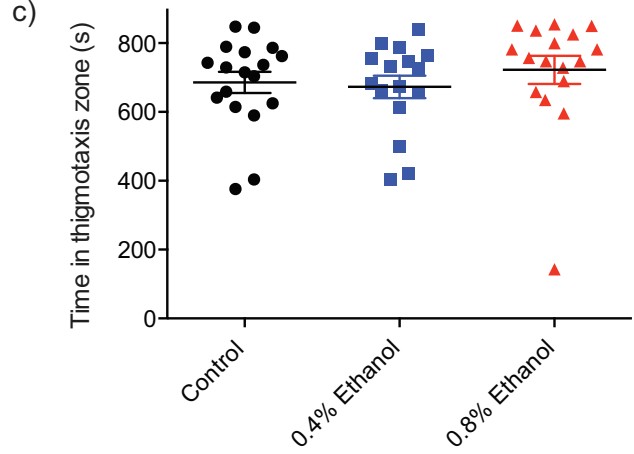

**Figure 2 Repeated ethanol administration effects on time in zones during withdrawal.** Each data point represents an individual fish. (A) Time in centre zone is decreased with 0.8% ethanol. (B) No effect on time spent in transition zone. (C) No effect on time spent in thigmotaxis zone. $^*P < 0.05$. Mean ± S.E.M.

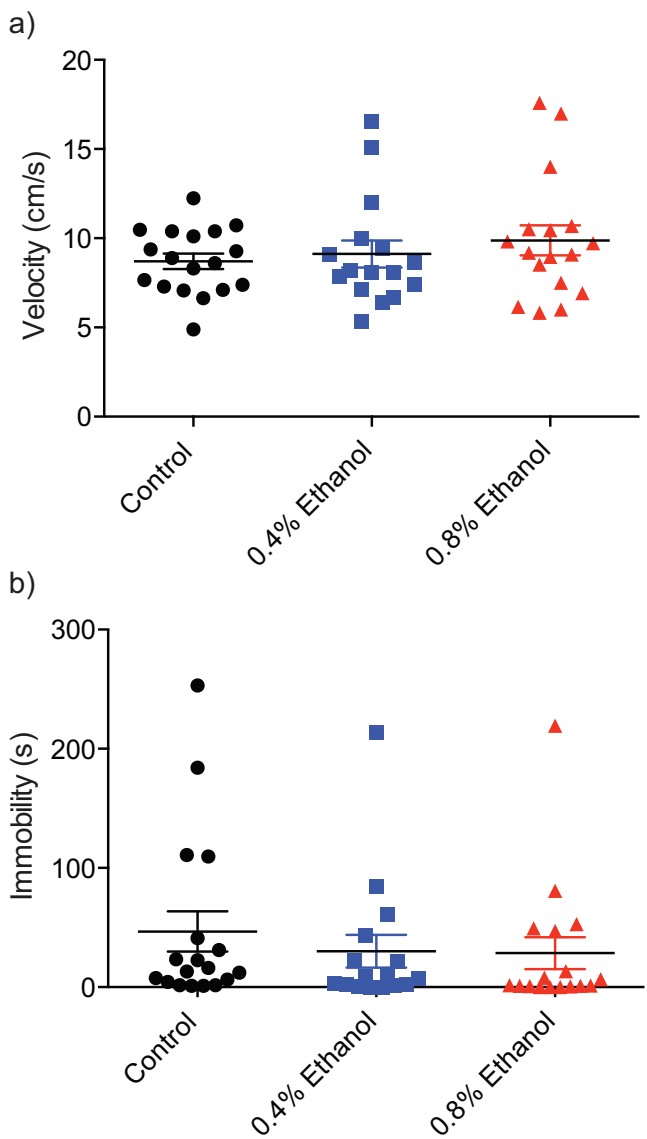

**Figure 3  Repeated ethanol administration effects on locomotion during withdrawal.** Each data point represents an individual fish. (A) No effect on velocity. (B) No effect on immobility. Mean ± S.E.M.

Specifically, during withdrawal from repeated daily doses of 0.8% ethanol, the time spent near the novel object decreased significantly compared to controls, indicative of an anxiogenic response (Fig. 2A).

To better understand the mechanisms behind the neuroadaptations caused by repeated ethanol administration, the use of animal models is essential, and the zebrafish model is becoming increasingly popular. To date, the analysis of anxiety-like behaviour after repeated or chronic ethanol exposure has been described (*Mathur & Guo, 2011*; *Tran, Chatterjee & Gerlai, 2015*; *Pittman & Hylton, 2015*; *Cachat et al., 2010*; *Pittman & Ichikawa, 2013*; *Gerlai et al., 2009*; *Müller et al., 2017*; *Holcombe et al., 2013*; *Holcombe, Schalomon & Hamilton, 2014*). Notably, *Mathur & Guo (2011)* used 20-min doses of

1% ethanol for 8 days, then used the novel tank diving test and the light/dark choice test to examine anxiety-like behaviours. Interestingly, in this study the anxiety-like behaviours were not observed in the first or second day of withdrawal, whereas we observed significant changes in anxiety-like behaviours within 2 days of withdrawal. However, *Mathur & Guo (2011)* found that after 6 days of withdrawal there was decreased bottom dwelling time in the novel tank diving test and increased dark preference in the light/dark test after 7 days of withdrawal; both indications of increased anxiety (*Mathur & Guo, 2011*). It is likely that we observed anxiogenic effects earlier (at 2 days of withdrawal) due to the greater length of daily exposure (i.e. −60 min as opposed to 20) and/or the duration of repeated exposures (i.e. −21 days vs 8 days). Another possible explanation is the difference in strain of zebrafish used; AB strain (*Mathur & Guo, 2011*) compared to wild-type in our study and others (*Rosemberg et al., 2012*; *Mocelin et al., 2018*; *Müller et al., 2017*). After 8 days of exposure to 1% ethanol for 20 min per day, social behaviour (inter-fish distance and farthest neighbour distance) was decreased on day 9, consistent with a tighter shoal and increased anxiety. Interestingly, this experiment was performed on wild-type zebrafish of mixed genders, much like the present study, and may indicate this heterogeneous genetic lineage undergoes withdrawal quicker than the AB strain (*Mathur & Guo, 2011*).

The candidate mechanism to explain how withdrawal from ethanol exposure is related to anxiety-like behaviour is neurochemical changes involving the GABAergic system, specifically GABA$_A$ receptors, resulting in increased behavioural excitability (*Gilpin & Koob, 2008*). Due to the constant presence of ethanol activating GABA$_A$ receptors, the mammalian nervous system compensates, altering the action of GABA-induced hyperpolarization by changing the subunit composition (*Cagetti et al., 2003*) and function (*Kang, Spigelman & Olsen, 1998*) of GABA$_A$ receptors. During ethanol withdrawal, there is less inhibition of excitatory activity by GABA$_A$ receptors compared to the basal state, leading to a hyperexcitable CNS (*Cagetti et al., 2003*). Therefore, normal levels of stimulation are likely to cause over-excitation due to the reduced suppression of the CNS, and results in the anxiogenesis observed during withdrawal. In the rodent chronic intermittent ethanol model, increased activity and anxiety is observed (*Cagetti et al., 2003*), similar to increased anxiety-like behaviour in zebrafish studies (*Da Silva Chaves et al., 2018*). The contribution of the GABAergic system to the increased anxiety caused by ethanol withdrawal in zebrafish is less clear. *Mahabir, Chatterjee & Gerlai (2018)* have recently reported no changes in neurochemicals (glutamate, GABA, aspartate, glycine, taurine) up to 102 days post fertilization after short embryonic ethanol exposure (2 h), whereas reductions of dopamine, 3,4-Dihydroxyphenylacetic acid (DOPAC), and serotonin have been observed in the AB but not TU strain (*Mahabir, Chatterjee & Gerlai, 2014*). This again highlights strain-specific differences and suggests future studies should examine neurochemical changes in wild-type zebrafish after repeated ethanol exposure.

Another possible mechanism for the anxiogenic state induced by repeated ethanol exposure in zebrafish is an increase in oxidative stress in the brain. As described above, repeated ethanol exposure for 8 days (20 min per day), results in altered social behaviour after 1 day of withdrawal (*Müller et al., 2017*). This coincides with biochemical

changes in the brain, including the decrease in superoxide dismutase (SOD) and catalase (CAT) enzymatic activity, leading to increased reactive oxygen species and potential oxidative damage via increased lipid peroxidation (*Müller et al., 2017*). Consistent with this is a study by *Agostini et al. (2018)* that found increasing ROS (CAT/SOD ratio) at 7, 14, and 28 days of ethanol (0.5%) exposure compared to controls. In our study, exposure to the lower concentration of 0.4% caused no behavioural changes, possibly due to less oxidative stress (below the threshold necessary to induce behavioural changes), resulting in behaviour similar to control fish. However, the previous statement is speculative as no ROS were measured, and needs experimental validation in order to be confirmed. Future studies should investigate a dose-response relationship between repeated ethanol administration, anxiety-like behaviour during withdrawal, and resultant oxidative stress.

Our study further demonstrates that the novel object approach test is an appropriate test to measure anxiety-like behaviour due to ethanol withdrawal. We have previously validated the use of the novel object approach test to examine anxiety-like behaviour with acute 1% ethanol doses, which resulted in zebrafish spending less time in the outer thigmotaxis zone and more time near the novel object in the transition zone (*Johnson & Hamilton, 2017*) as well as increased time in the centre zone near the object (*Hamilton et al., 2017*), indicative of decreased anxiety. Here, we have shown that repeated ethanol exposure produces the opposite response during withdrawal—decreased time spent near the centre object—indicative of increased anxiety.

The novel tank diving test, light/dark preference test, predator avoidance test, and shoaling test have also demonstrated increased anxiety-like behaviour following ethanol withdrawal (reviewed in *Da Silva Chaves et al., 2018*), supporting the findings reported here. This contrasts our previous finding that time in the light zone was increased in the light/dark test with 21-days of repeated daily ethanol exposure (0.2%; *Holcombe et al., 2013*). However, in this previous study (*Holcombe et al., 2013*) we suggest that since the fish were dosed with white walls surrounding their dosing tank, we had induced a conditioned-place preference for the white zone of the light/dark test, resulting in a preference for the white side of the light/dark test (*Holcombe et al., 2013*). The current study supports this hypothesis, since we observed increased anxiety-like behaviour using a different test, the novel object approach test, under the similar conditions. Although this should be experimentally tested, it suggests that careful consideration of the visual surroundings with repeated administration studies and the use non-relevant visual cues when using the light/dark test is essential to properly assess withdrawal-induced anxiety-like behaviour.

The present findings demonstrate an effect of repeated intermittent ethanol administration-induced withdrawal and anxiogenic behavioural changes. Consistent with other pharmacological studies examining zebrafish behaviour, our study supports the use of zebrafish as an effective model organism for studying drug-induced withdrawal. Additionally, we have found further support for the novel object approach test as an appropriate test for examining anxiety-like behaviours in zebrafish. Overall, there is great translational relevance in studying the anxiogenic effects of withdrawal due to repeated ethanol exposure in zebrafish.

### Funding

This work was supported by the Natural Sciences and Engineering Research Council of Canada (NSERC) Discovery grants to Trevor J. Hamilton (No. 04843) and Matthew S. Ross (No. 04852). The funders had no role in study design, data collection and analysis, decision to publish, or preparation of the manuscript.

### Grant Disclosure

The following grant information was disclosed by the authors:
Natural Sciences and Engineering Research Council of Canada, NSERC: 04843 and 04852.

### Competing Interests

The authors declare that they have no competing interests.

### Author Contributions

- Jeffrey T. Krook conceived and designed the experiments, performed the experiments, analysed the data, prepared figures and/or tables, authored or reviewed drafts of the paper, approved the final draft.
- Erika Duperreault performed the experiments, approved the final draft.
- Dustin Newton performed the experiments, approved the final draft.
- Matthew S. Ross conceived and designed the experiments, authored or reviewed drafts of the paper, approved the final draft.
- Trevor J. Hamilton conceived and designed the experiments, analysed the data, contributed reagents/materials/analysis tools, prepared figures and/or tables, authored or reviewed drafts of the paper, approved the final draft.

### Animal Ethics

The following information was supplied relating to ethical approvals (i.e. approving body and any reference numbers):

All experiments were approved by the MacEwan University Animal Research Ethics Board (AREB) under protocol number 05-12-13, in compliance with the Canadian Council for Animal Care (CCAC) guidelines for the care and use of experimental animals.

### Data Availability

The raw data are available in the Supplementary File.

### Supplemental Information

Supplemental information for this article can be found online at http://dx.doi.org/10.7717/peerj.6551#supplemental-information.

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
