# Peer review of "Repeated ethanol exposure increases anxiety-like behaviour in zebrafish during withdrawal"

_PeerJ, doi:10.7717/peerj.6551_

## Round 0.1 · original submission · Major Revisions

The reviewers prepared several comments and queries for your consideration as you can see below. I believe that all comments will improve your manuscript quality. I suggest that the form of randomization and blinding be explicitly described in the paper to enable replication by other authors. I, therefore, invite you to revise and resubmit your manuscript, considering the points above as well as the issues raised by the reviewers.

·

Basic reporting

The manuscript by Krook et al. reports on the behavioral effects of ethanol withdrawal on zebrafish behavior in the novel object test (to which the authors refer as a “novel approach test”, without much consequence). They find that zebrafish exposed to 0.8% EtOH, but not 0.4% EtOH, for 21 days, with an exposure of 1 h per day, and later subjected to 48 h withdrawal, appear to either avoid the novel object or display more centrophobia, without motor impairments. Results are discussed straightforwardly in terms of behavioral effects of ethanol withdrawal, as well as of possible neurochemical mechanisms.
The paper introduces a straightforward and relatively selective result in a field that is marked by contradictory findings. Authors introduce a novel approach to study the effects of ethanol withdrawal, building on previous research to test behavioral hypothesis. Although no justification is given for the sample sizes used, they appear appropriate.

Experimental design

1) There are several issues of experimental design and transparency which make it difficult to understand biases that could have been introduced in the experiment and would be responsible for some of the anomalous findings. These include:
-How were sample sizes calculated? Based on what effect size, power, and alpha? This is crucial to understand whether experiments are under-powered.
-Were experimenters blind to treatment? We know that animals were randomly allocated (lines 129), but the specifics are not reported. Be specific and explain HOW blinding and random allocation was performed.
-Were animals from the control and treatment groups drawn from the same tank/rack? If not, were the racks/tanks positioned at different places in the vivarium?
-Were only males used? Only females? Mixed sexes? What was the phenotype of the fish (e.g., shortfin, longfin)? What was the origin of the fish before they reached the colony? Were they bred in the lab? For how many generations?
-At what time of the day did experiments take place?
2) The dataset provided is formatted in a way that precludes computational reproducibility, as one needs to either format it or copy-paste to another program to make analyses. Please provide data formatted as comma-separated values spreadsheet (CSV), with machine-readable headers (i.e., using dots or underlines as separators), and six columns (group, time in center zone, time in transition zone, time in thigmotaxis zone). Also, please provide raw data for velocity and immobility.

Validity of the findings

The issues of transparency and design described in the previous session need to be addressed in order to judge the validity of the findings.

Additional comments

1) In the results, data are presented as means ± SOMETHING; are those CIs, S.E.M.s, or SDs?
2) While the name “novel approach test” is interesting, for the sake of comparison with results from other labs I suggest changing it back to “novel object test”
3) Please provide a full protocol for the behavioral test (it can be posted on protocols.io, e.g.)

·

Basic reporting

The manuscript depicts the anxiogenic effect of ethanol during withdrawal. Despite the relevance of the study and consider the manuscript to be publishable, however, the presented effect is not supported in literature studies, as mentioned in the manuscript. The concentrations and time of exposure used by the authors do not match those of the literature (see review), therefore, I consider that the observed behavioral effect should also be reproduced in other behavioral protocols. I consider that the breadth of conclusions requires additional experiments. The same occurs when affirming a dose-dependent effect using only 2 concentrations of ethanol, what are the behavioral effects of higher and lower concentrations of those used in the study?

1) Authors begin and end phrases throughout the manuscript without demonstrating the reference.

2) In lines 59-63 the authors describe zebrafish positives on genetic manipulation and model neurotransmitter systems provoked by ethanol, however, the reference used (Müller et al., 2017) carried out studies of social behavior and oxidative stress, not those described in the manuscript.

3) The data presented by the authors show a possible anxiogenic effect of ethanol, however, the concentrations and time of exposure are not the same presented in the literature, referenced and discussed by the authors. I consider that a behavioral protocol is not adequate to affirm the presented effect, since there is no data in the literature that use 21 days of exposure and concentrations of 0.4% and 0.8% therefore, additional studies are necessary to confirm the anxiogenic effect.

4) In lines 59-63 the authors describe genetic manipulation and neurotransmitter systems induced by ethanol in zebrafish, although the reference used (Müller et al., 2017) carried out studies of social behavior and oxidative stress, not those described. This type of error also occurs in lines 134-135, 227-230, among others. I suggest that the authors review it.

Experimental design

1) The manuscript has scientific relevance, however, the data presented in only one behavioral apparatus are not considered enough to confirm the observed effect. I consider the need to include at least one more behavioral protocol proving such an effect, as done by Mathur & Guo (2011) since the concentrations and time of exposure chosen by the authors are not the same as those found in the literature.

2) The authors mention that Mathur & Guo (2011) do not observe effects on day 1 and 2 after withdrawal in the tests of light/dark and novel tank test, respectively. However, Müller et al. (2017) presented an anxiogenic effect in the social behavior test, also correlating oxidative damage induced by ethanol within 24 hours after withdrawal. Why did the authors opt to perform the experiments only 2 days later?

3) Were the experiments performed only with male animals? or females? To describe in the materials and methods.

4) Were the animals fed only once a day (line 120)?

5) What was the procedure after behavioral analysis? Did the animals return to the dwelling tank? Were they euthanized?

6) In the description of the exposure protocol, the authors reported their concentrations and exposure times according to Holcombe et al., 2013 (30 min exposure in 0.2% and 1.4% ethanol) and Mathur & Guo 2011 (20 min exposure in 1% of ethanol), in addition to describing that the exposure time was considered 60 min according to a study by Tran, Chatterjee, Gerlai, 2016 (which used 30 min exposure in the protocol). Why did the authors choose to use 0.4% and 0.8% of ethanol for 21 days, since the study on which the time and concentration of exposure was based does not correspond to of the study?

7) How was the preparation of the exposure tanks, both ethanol, and control? What water was used?

8) I suggest that the authors add an experimental design showing the arena and the LEGO figurine used, as done in the works cited (Ou et al., 2015, Hamilton et al., 2017).

9) The behavioral analysis was performed for 5 min, however, the studies with this behavioral protocol do the analysis for 10 min. Why did the authors opt for this time?

10) Why did the authors decide to exclude animals with more than 300 seconds of immobility from the test?

Validity of the findings

1) In the time results in the zones, there was no difference between the groups 0.4% and 0.8%? If the authors say it is dose-dependent, is there no need to show that significant difference?

2) It lacked to describe in detail what the principle of the behavioral protocol chosen and its effect on the zebrafish. Is not clear the use of LEGO in the protocol, what his role in the test? How does the zebrafish behave with and without the presence of the figurine?

3) In lines 252-256, the authors state that low concentrations (0.4%) result in less oxidative stress and, for this reason, the group behaves similarly to the control group. Have the authors evaluated the oxidative stress in the study? How can the authors conclude this statement?

4) A recent study (Agostini et al., 2018, doi: 10.1007 / s12640-017-9816-8) showed exposure to 0.5% ethanol for 7, 14 and 28 days and concluded that the longer the exposure time of the animals to ethanol, the lower the oxidative damage (in 28 days no lipid peroxidation occurs, ROS levels are present only in 7 days of exposure). Considering the difference of 0.1% ethanol between the manuscript and Agostini's study (2018), can the authors conclude that there was less oxidative stress?
5) The authors conclude that there is a dose-dependent relationship with ethanol, but this was not evident in the graphs at the same time, the concentration curve should be higher for such a conclusion.

6) During the discussion, the authors mention results from the literature that differ from those found in the manuscript; however, the concentration, exposure time, zebrafish strains and behavioral devices are also different from the present study. For example, between the lines 258-265, the authors describe that the results presented in the manuscript are the opposite of those in the literature, however, reinforcing again the idea that the concentrations (0.2%, 1%, and 1.4%) of the studies beyond the exposure time (30 min) produce fewer anxiety effects. Studies of Rosemberg et al., 2012 (doi: 10.1016 / j.neuropharm.2012.05.009) and Mocelin et al., 2018 (doi: 10.1007 / s11064-017-2442-2) demonstrate opposite effects (anxiogenic in adult zebrafish) on acute exposure for 60 min. A new literature review could support the data found in the manuscript (anxiogenic effect), perhaps demonstrating that even at lower concentrations of exposure and a greater number of days, ethanol is capable of promoting an anxiogenic effect.

Additional comments

1) Line 46-47: Which membranes are activated?

2) Line 54-57: It is confused as described.

3) Line 71-99: This extensive and confusing. The authors address behavioral issues after mentioning biochemical results. There is no connection between sentences. Rewrite.

4) Line 78-80: What did the study cited bring about? What is the relation with the study of the authors?

5) Line 91: Are the authors referring to the study of Mathur & Guo 2011?

6) Line 145: behavioural tests or test?

7) Line 220-222: Only concentrations of 1.0% and 1.5% of ethanol were used in the study cited? And the 0.5% concentration that increased time on the light side in the light/dark test?
8) Line 227: Set reference.

9) Line 227-230: The study by da Silva Chaves et al. (2018) performs continuous exposure with withdrawals of 60 minutes (in water) for 16 days, and not described in the sentence, is confusing.

10) Line 240-244: The authors write about the GABAergic system and finally bring studies in rodents emphasizing alteration of sensitivity to hypnotic drugs and steroids, not following sentence line of thought. Do authors find it better to remove?

11) I do not think it is appropriate to use the term dose-dependent in the abstract and in the text since the number of concentrations used is insufficient for the protocol used. Can you state the behavior of the animals in smaller and larger concentrations of ethanol?

12) Provide missing references in the text (lines 30, 32, 37, 47, 50-52, 85, 213-215, 240, 242).

13) Add the parameters analyzed by the authors in line 105.

·

Basic reporting

The manuscript is written in good english and with clarity in the presentation of ideas and data from literature. However, the abstract is poor in details. The structure, figures and raw data were presented correctly.
The issue of the manuscript is of obvious importance, since the neurochemical and behavioral aspects of alcohol withdrawal are far from clear, whereas the health and economic burden has enormous impacts around the world. However, the introduction of this manuscript does not cover the main aspects of ethanol withdrawal (eg, dopamine, serotonin, corticotropin-releasing factor levels, etc.) and the recent contribution of zebrafish studies to this issue (eg: Chaves et al., 2018 , Benneh et al., 2017).

Experimental design

The main aim of the manuscript is interesting on the basis of the heterogeneity found in the data available about this issue in the literature. The methodology, while presented as a new approach to test anxiety-like behaviour in the context of ethanol withdrawal, do not add strongest contributions to the area. Some minors experimental points also contributes to the weakness of the manuscript:
(1) the authors discussed the data based in a dose-dependent relationship between behaviour and ethanol withdrawal, but only two doses were tested and just the higher had effect, which is not enought to determine a dose-dependence of factors (Discussion: 209-211; Conclusion: lines 281-282)
(2) The validity of the test should be checked by an anxiolytic drug.
(3)The title of the manuscript enphasizes "repeated ethanol exposure increase anxiety-like behaviour....", however the methodology do not permit to compare to any anxiety-like effect of ethanol.
(4) Also, the age, weight of animals were not detailed. The light/dark period of 12/12 is not common and should be referenced.

Validity of the findings

While the assumption is of interest the results are too preliminary to add novelty to this area. Only one behavioural parameter was altered and no neurochemical aspect was considered (cortisol levels, dopamine, serotonin, glutamate levels, GABA receptor expression,....). The discussion is based in the explanation of how withdrawal occurs, but the experimental design is not focused on that, and any kind of behavioural test could theoretically discuss in this same way even if the methodolgical approach was different.

Additional comments

No additional comment

---

## Round 0.2 · Minor Revisions

Reviewer 2 provided minor comments and queries for your attention, and I their report is below. I believe that all comments will improve your manuscript quality. I invite you to revise and resubmit your manuscript, considering the points raised.

·

Basic reporting

no comment

Experimental design

no comment

Validity of the findings

no comment

Additional comments

Dear Editor.

The authors responded to the review comments, however, some doubts remain unclear. Problems were also observed in the manuscript and described in the review below.

Initially, disregarding future experiments, I believe that a review has been made in the literature regarding ethanol concentrations, their respective behavioral responses in other protocols, for example using the same 21 days of exposure, I continue to question why use a range of concentrations and not use what is already in the literature?

I suggest that it be described in which animal model these effects were observed, and if this model was withdrawn, intermittent, chronic. And what about zebrafish? and in humans?

Materials and methods:

The authors describe that daily exposure was performed between 8 A.M. and 4 P.M. I suggest describing how this exposure was made, how many fish per tank, in what proportion of males and females?

How were these animals fed? It the authors feed the animals once a day (not described in what time), are there groups received food and were exposed to ethanol at different time intervals, do not the authors consider a possible limitation of the study?

In the preparation of ethanol at the tank, the animals were transferred before or after the add the ethanol?

The authors added in the manuscript that animals of both sexes were used, however, it was not clear the proportion of males and females per group and how these animals were previously housed.

Already described in the description item of the exposure protocol that researchers were not blind to the treatment groups during dosing or testing, I suggest you remove the description in the item of Behavioural testing.

Discussion:

What were the neural changes that occurred? What's the reference?

Line 235-239: The authors did not evaluate neural changes in the experiment, this description is more introductory than discursive. I suggest you remove it from the manuscript.

Line 247-252: The authors describe that the study of Rosemberg (2012) and Mocelin (2018) observed an anxiogenic effect in 20 min of exposure, however in 20 min the effects are anxiolytic (Rosemberg 2012) and were not the time used by Mocelin (2018). It should also be considered that the protocols mentioned above are of acute, not chronic exposure of the authors Mathur and Guo (2011). This discussion needs to be reviewed and best described.

Line 251: Rewrite reference: Rosemberg, not Rosemburg.
The authors should consider that the animal model used by Mathur and Guo (2011) is AB, different from those used in non-study and studies by Rosemberg (2012) and Mocelin (2018) (wild-type). This is an important point to be evaluated since studies show behavioral and neurochemical differences between zebrafish strains.

Using wild-type lineage may be considered more heterogeneous and best represents the high rate of genetic variability, as in humans. I suggest a discussion about the theme regarding the events observed by the authors and the withdrawal model of Mathur and Guo (2011) and Muller (2017).

doi: 10.1186/1471-2202-11-90
doi: 10.1007/s00726-013-1658-y
doi: 10.1016/j.ntt.2018.05.005

Have no studies on zebrafish and rodents assessed the activity of GABA receptors? What do the authors consider similar in zebrafish, the activity of GABA?
- See Mahabir (2018) 10.1016/j.ntt.2018.05.005

Since the authors describe that the ROS data in the discussion are merely speculative, I suggest that the authors also report in this way the possible involvement of the GABA system.

·

Basic reporting

The overall manuscript was improved. While I am still concerned about the validity of the behavioural test for anxiety, the authors convinced me that the literature present similar works where the anxiety parameters were checked using similar approach.

Experimental design

The experimental design is now precisely detailed.

Validity of the findings

Based on data from this work and literature, the conclusion of the manuscript is acceptable.

Additional comments

The manuscript was improved and probably will encourage further investigation especially on neurochemical aspects of ethanol-withdrawal.

---

## Round 0.3 · accepted · Accept

I am pleased to inform you that your manuscript referenced above has been accepted for publication in PeerJ. Good job!

#